# Generalized Information Bottleneck for Gaussian Variables

## Abstract

The information bottleneck (IB) method offers an attractive framework for understanding representation learning, however its applications are often limited by its computational intractability. Analytical characterization of the IB method is not only of practical interest, but it can also lead to new insights into learning phenomena. Here we consider two different generalizations of the IB problem, in which the mutual information is replaced by correlation measures based on Rényi and Jeffreys divergences, respectively. We derive an exact, analytical IB optimal linear Gaussian encoder for Gaussian correlated variables. Our analysis reveals a series of structural transitions, similar to those previously observed in the original IB case. We find further that although solving the original, Rényi and Jeffreys IB problems yields different representations in general, the structural transitions occur at the same critical tradeoff parameters, and the Rényi and Jeffreys IB solutions perform well under the original IB objective. Our results suggest that formulating the IB method with alternative correlation measures could offer a strategy for obtaining an approximate solution to the original IB problem.

## 1 Introduction

Effective representation of data is key to generalizable learning. Characterizing what makes such representation good and how it emerges is crucial to understanding the success of modern machine learning. The information bottleneck (IB) method—an information-theoretic formulation for representation learning (Tishby et al., 1999)—has proved a particularly useful conceptual framework for this question, and has led to a deeper understanding of representation learning in both supervised and self-supervised learning (Achille & Soatto, 2018a;b; Tian et al., 2020; Zbontar et al., 2021). Investigating this abstraction of representation learning has the potential to yield new insights that are applicable to learning problems.

Quantifying the goodness of a representation requires the knowledge of what is to be learned from data. Information bottleneck theory exploits the fact that, in many settings, we can define relevant information through an additional variable; for example, it could be the label of each image in a classification task. This notion of relevance allows for a precise definition of optimality—an IB optimal representation $T$ is maximally predictive of the relevance variable $Y$ while minimizing the number of bits extracted from the data $X$. The IB method formulates this principle as an optimization problem (Tishby et al., 1999),

$$\min_{Q_{T|X}} \ \mathrm{IB}_\beta(Q_{T|X}; P_{XY}) \quad \text{with} \quad \mathrm{IB}_\beta(Q_{T|X}; P_{XY}) = I(T; X) - \beta I(T; Y). \tag{1}$$

Here the optimization is over the encoders $Q_{T|X}$ which provide a (stochastic) mapping from $X$ to $T$. Maximizing the mutual information $I(T; Y)$ [second term in Eq (1)] encourages a representation $T$ to encode more relevant information while minimizing $I(T; X)$ [first term in Eq (1)] discourages it from encoding irrelevant bits. The parameter $\beta > 0$ controls the fundamental tradeoff between the two information terms.

The IB method has proved successful in a number of applications, including neural coding (Palmer et al., 2015; Wang et al., 2017; 2021), statistical physics (Still et al., 2012; Gordon et al., 2021; Kline & Palmer, 2022), clustering (Strouse & Schwab, 2019), deep learning (Alemi et al., 2017; Achille & Soatto, 2018a;b), reinforcement learning (Goyal et al., 2019) and learning theory (Bialek et al., 2001; Shamir et al., 2010; Bialek et al., 2020; Ngampruetikorn & Schwab, 2022). However the nonlinear nature of the IB problem makes it

computationally costly. Although scalable learning methods based on the IB principle are possible thanks to variational bounds of mutual information (Alemi et al., 2017; Chalk et al., 2016; Poole et al., 2019), the choice of such bounds, as well as specific details on their implementations, can introduce strong inductive bias that competes with the original objective (Tschannen et al., 2020).

While large-scale applications of the IB method in its exact form are generally intractable, special cases exist. For example, the limit of low information—i.e., when both terms in Eq (1) are small—can be described by a perturbation theory, which provides a recipe for identifying a representation that yields maximum relevant information per extracted bit (Wu et al., 2019; Ngampruetikorn & Schwab, 2021). But perhaps the most important special case is when the source $X$ and target $Y$ are Gaussian correlated random variables (see Sec 2.3). In this case, an exact *analytical* solution exists (Chechik et al., 2005). Despite the seemingly restrictive Gaussian assumption, this result has proved useful in practice. In particular, it is possible to control the data in many interesting settings. The exact IB solution allows for a principled and exact investigation of the optimality and adaptability of learning systems, see, e.g., Palmer et al. (2015).

Although originally formulated with Shannon mutual information, the fundamental tradeoff in the IB method applies more generally: the IB optimization, Eq (1), can be formulated with other mutual dependence measures, see, e.g., Harremöes & Tishby (2007); Hsu et al. (2018). Here we consider generalized IB problems based on two important correlation measures. The first is a parametric generalization of Shannon information, based on Rényi divergence (Rényi, 1961). Rényi-based generalizations of mutual information and entropy are central in quantifying quantum entanglement (Horodecki et al., 2009; Eisert et al., 2010) and have proved a powerful tool in Monte-Carlo simulations (Hastings et al., 2010; Singh et al., 2011; Herdman et al., 2017) as well as in experiments (Islam et al., 2015; Bergschneider et al., 2019; Brydges et al., 2019). The second mutual dependence measure we consider is based on Jeffreys divergence (Jeffreys, 1946). The resulting Jeffreys information is (up to a constant prefactor) equal to the generalization gap of a broad family of learning algorithms, known as Gibbs algorithms (Aminian et al., 2021).

We specialize to the case of Gaussian correlated variables, and derive an analytical solution for two separate generalizations of the IB problem: the Rényi and Jeffrey cases in §3 and §4, respectively. Our derivations extend the result of Chechik et al. (2005) to a class of information-theoretic mutual dependence measures which includes Shannon information as a limiting case. We show that, for both Rényi and Jeffreys cases, an optimal linear Gaussian encoder can be constructed from the eigenmodes of the normalized regression matrix $\Sigma_{X|Y}\Sigma_X^{-1}$. Our solution reveals a series of phase transitions, similar to those observed in the Gaussian IB method (Chechik et al., 2005). In both Rényi and Jeffreys cases, we find that although the optimal encoders depend on information measures, the phase transitions occur at the critical tradeoff parameters $\beta_c^{(i)}$ that coincide with that of the Shannon case, independent of the order of Rényi information.

## 2 Background

### 2.1 Divergence-based Correlation measure

When two random variables $X$ and $Y$ are uncorrelated, their joint distribution $P_{XY}$ is equal to the product of their marginals $P_X$ and $P_Y$. As a result, we can quantify the mutual dependence between $X$ and $Y$ by the difference between $P_{XY}$ and $P_X \otimes P_Y$,

$$\Omega(X;Y) \equiv \mathcal{D}(P_{XY} \parallel P_X \otimes P_Y). \tag{2}$$

Here $\mathcal{D}(P \parallel Q)$ denotes a statistical divergence which, by definition, is nonnegative and vanishes if and only if $P = Q$. When defined with the Kullback–Leibler (KL) divergence, the above measure becomes Shannon information, $I(X;Y) = D_{\mathrm{KL}}(P_{XY} \parallel P_X \otimes P_Y)$.

**Definition 2.1.** *The generalized IB problem is characterized by the loss function*

$$\mathrm{IB}_\beta^{(\Omega)}(Q_{T|X}; P_{XY}) = \Omega(T;X) - \beta\Omega(T;Y),$$

*where $\Omega(\,\cdot\,;\,\cdot\,)$ is a measure of statistical correlation between two random variables.*

Below we use Rényi and Jeffrey divergences to provide concrete definitions for Rényi $q$–information and Jeffrey information, respectively.

### 2.1.1 Rényi $q$–information

**Definition 2.2.** *We define* Rényi $q$-information *as*

$$I_q(X;Y) \equiv \mathcal{R}_q(P_{XY} \parallel P_X \otimes P_Y) \quad for \quad q \in (0,1) \cup (1,\infty), \tag{3}$$

*where $\mathcal{R}_q$ denotes Rényi divergence of order $q$ (Rényi, 1961),*

$$\mathcal{R}_q(P \parallel Q) = \frac{1}{q-1} \ln \int dQ \left( \frac{dP}{dQ} \right)^q. \tag{4}$$

This definition extends to $q = 0, 1$ and $\infty$ via continuity in $q$. In particular, $\mathcal{R}_1(P\|Q) = D_{\mathrm{KL}}(P\|Q)$ (van Erven & Harremoës, 2014, Thm 5), and as a result $I_1(X;Y) = I(X;Y)$. Rényi divergences, and thus $q$-information, satisfy the data processing inequality since they have a strictly increasing relationship with an $f$-divergence [with $f(t) = (t^q - 1)/(q-1)$] which exhibits this property, see, e.g., Liese & Vajda (2006).

**Lemma 2.1.** *Let $\begin{bmatrix} X \\ Y \end{bmatrix} \sim \mathcal{N}(\begin{bmatrix} \mu_X \\ \mu_Y \end{bmatrix}, \begin{bmatrix} \Sigma_X & \Sigma_{XY} \\ \Sigma_{YX} & \Sigma_Y \end{bmatrix})$. Rényi information is given by (see Appendix B for derivation)*

$$I_q(X;Y) = -\frac{1}{2\bar{q}} \ln \frac{|\Sigma_{X|Y}\Sigma_X^{-1}|^{\bar{q}}}{|I - \bar{q}^2(I - \Sigma_{X|Y}\Sigma_X^{-1})|} \quad with \quad \bar{q} = 1 - q, \tag{5}$$

*where $I$ denotes the identity matrix and $|M|$ the determinant of a matrix $M$.*

We see that this information depends on the covariance matrices only through the normalized regression matrix $\Sigma_{X|Y}\Sigma_X^{-1}$. We note also that this information can diverge when $q > 2$ since the eigenvalues of $\Sigma_{X|Y}\Sigma_X^{-1}$ range from zero to one (Chechik et al., 2005, Lemma B.1). It is easy to verify that Shannon information corresponds to the limit $q \to 1$,

$$I(X;Y) = \lim_{q \to 1} I_q(X;Y) = -\frac{1}{2} \ln |\Sigma_{X|Y}\Sigma_X^{-1}|. \tag{6}$$

In addition, we note that for Gaussian variables $I_2(X;Y) = 2I(X;Y)$ and $I_q(X;Y)$ increases with $q$ from zero at $q = 0$.

Note that alternative definitions of Rényi mutual information exist. In physics literature, a frequently used definition is $\tilde{I}_q(X;Y) = S_q(X) + S_q(Y) - S_q(X,Y)$ where $S_q(X) = (1-q)^{-1} \ln \int dx \, p_X(x)^q$ is Rényi (differential) entropy of order $q$ (see, e.g., Brydges et al. (2019, Eq 4)). However, for Gaussian variables, this definition leads to Rényi information that is equal to Shannon information regardless of $q$ (see Appendix B.1).

### 2.1.2 Jeffrey information

**Definition 2.3.** *We define Jeffreys information by*

$$J(X;Y) \equiv D_{\mathrm{J}}(P_{XY} \parallel P_X \otimes P_Y), \tag{7}$$

*where $D_{\mathrm{J}}$ is Jeffreys divergence (Jeffreys, 1946),*

$$D_{\mathrm{J}}(P \parallel Q) = \frac{1}{2}[D_{\mathrm{KL}}(P \parallel Q) + D_{\mathrm{KL}}(Q \parallel P)]. \tag{8}$$

Jeffreys divergence, and thus Jeffrey information, satisfies the data processing inequality since it is an $f$-divergence, see, e.g., Liese & Vajda (2006).

**Lemma 2.2.** *Let $\begin{bmatrix} X \\ Y \end{bmatrix} \sim \mathcal{N}(\begin{bmatrix} \mu_X \\ \mu_Y \end{bmatrix}, \begin{bmatrix} \Sigma_X & \Sigma_{XY} \\ \Sigma_{YX} & \Sigma_Y \end{bmatrix})$. Jeffreys information reads (see Appendix C)*

$$J(X;Y) = \frac{1}{2} \operatorname{tr} \left( \Sigma_X \Sigma_{X|Y}^{-1} - I \right). \tag{9}$$

Note that, unlike Shannon information, Jeffreys information is not a special case of Rényi information.

## 2.2 Canonical representation of Gaussian correlated variables

Consider Gaussian correlated variables

$$\begin{bmatrix} X \\ Y \end{bmatrix} \sim \mathcal{N}\left( \begin{bmatrix} \mu_X \\ \mu_Y \end{bmatrix}, \begin{bmatrix} \Sigma_X & \Sigma_{XY} \\ \Sigma_{YX} & \Sigma_Y \end{bmatrix} \right). \tag{10}$$

The canonical representation is a linear transformation, $\tilde{X} = K_X(X - \mu_X)$ and $\tilde{Y} = K_Y(Y - \mu_Y)$, which results in a factorizable joint distribution $P_{\tilde{X}\tilde{Y}} = \prod_i P_{\tilde{X}_i \tilde{Y}_i}$ where $P_{\tilde{X}_i \tilde{Y}_i}$ is a bivariate Gaussian distribution, characterized by unit variance $\sigma_{\tilde{X}_i} = \sigma_{\tilde{Y}_i} = 1$ and a correlation coefficient $\rho_i$. When the dimensions of $X$ and $Y$ are unequal, the joint distribution is a product of the bivariate Gaussian distributions for the matched dimensions and a standard Gaussian distribution for the unmatched dimensions.

**Lemma 2.3** [Globerson & Tishby (2004, §4)]. *For $\begin{bmatrix} X \\ Y \end{bmatrix} \sim \mathcal{N}(\begin{bmatrix} \mu_X \\ \mu_Y \end{bmatrix}, \begin{bmatrix} \Sigma_X & \Sigma_{XY} \\ \Sigma_{YX} & \Sigma_Y \end{bmatrix})$, let $V = [v_1 \; v_2 \; \cdots]^{\mathsf{T}}$ be a matrix of left (row) eigenvectors of $\Sigma_{X|Y}\Sigma_X^{-1}$ and $\Lambda = \mathrm{diag}(\lambda_1, \lambda_2, \cdots)$ the diagonal matrix of corresponding eigenvalues, (i.e., $V\Sigma_{X|Y}\Sigma_X^{-1} = \Lambda V$). Then, there exist invertible matrices, $K_X$ and $K_Y$, such that*

$$\begin{bmatrix} \tilde{X} \\ \tilde{Y} \end{bmatrix} = \begin{bmatrix} K_X(X - \mu_X) \\ K_Y(Y - \mu_Y) \end{bmatrix} \sim \mathcal{N}\left( 0, \begin{bmatrix} I & (I - \Lambda)^{\frac{1}{2}} I_{XY} \\ I_{YX}(I - \Lambda)^{\frac{1}{2}} & I \end{bmatrix} \right),$$

*where $[I_{XY}]_{ij} = \delta_{ij}$ is a diagonal rectangular matrix.*

The proof follows the logical steps in Globerson & Tishby (2004).

*Proof.* We will use three identities (see, e.g., Chechik et al. (2005)):

    (a) Let $R = V\Sigma_X V^{\mathsf{T}}$. Then, $R$ is diagonal,
    (b) $V\Sigma_{X|Y}V^{\mathsf{T}} = \Lambda R$,
    (c) $V\Sigma_{XY}\Sigma_Y^{-1}\Sigma_{YX}V^{\mathsf{T}} = (I - \Lambda)R$.

We start with the proof of (a). Multiplying $V\Sigma_{X|Y}\Sigma_X^{-1} = \Lambda V$ by $\Sigma_X^{\frac{1}{2}}$ from the right and rearranging yields $(V\Sigma_X^{\frac{1}{2}})\Sigma_X^{-\frac{1}{2}}\Sigma_{X|Y}\Sigma_X^{-\frac{1}{2}} = \Lambda(V\Sigma_X^{\frac{1}{2}})$, which shows that $V\Sigma_X^{\frac{1}{2}}$ is a matrix of eigenvectors of a symmetric matrix $\Sigma_X^{-\frac{1}{2}}\Sigma_{X|Y}\Sigma_X^{-\frac{1}{2}}$. Therefore $V\Sigma_X^{\frac{1}{2}}$ is an orthogonal matrix and $V\Sigma_X^{\frac{1}{2}}(V\Sigma_X^{\frac{1}{2}})^{\mathsf{T}} = V\Sigma_X V^{\mathsf{T}} = R$ is diagonal. Next, we show that (b) and (c) follow from direct algebraic manipulations: $V\Sigma_{X|Y}V^{\mathsf{T}} = V\Sigma_{X|Y}\Sigma_X^{-1}\Sigma_X V^{\mathsf{T}} = \Lambda V\Sigma_X V^{\mathsf{T}} = \Lambda R$ and $V\Sigma_{XY}\Sigma_Y^{-1}\Sigma_{YX}V^{\mathsf{T}} = V(\Sigma_X - \Sigma_{X|Y})V^{\mathsf{T}} = (I - \Lambda)R$.

Suppose $\tilde{X} = R^{-\frac{1}{2}}VX$ and $\tilde{Y} = I_{YX}(I - \Lambda)^{-\frac{1}{2}}R^{-\frac{1}{2}}V\Sigma_{XY}\Sigma_Y^{-1}Y$. Then,

$$\Sigma_{\tilde{X}} = R^{-\frac{1}{2}}V\Sigma_X V^{\mathsf{T}}R^{-\frac{1}{2}} = R^{-\frac{1}{2}}RR^{-\frac{1}{2}} = I \tag{11}$$

$$\Sigma_{\tilde{Y}} = I_{YX}(I - \Lambda)^{-\frac{1}{2}}R^{-\frac{1}{2}}V\Sigma_{XY}\Sigma_Y^{-1}\Sigma_Y\Sigma_Y^{-1}\Sigma_{YX}V^{\mathsf{T}}(I - \Lambda)^{-\frac{1}{2}}R^{-\frac{1}{2}}I_{XY} \tag{12}$$

$$= I_{YX}(I - \Lambda)^{-\frac{1}{2}}R^{-\frac{1}{2}}V\Sigma_{XY}\Sigma_Y^{-1}\Sigma_{YX}V^{\mathsf{T}}R^{-\frac{1}{2}}(I - \Lambda)^{-\frac{1}{2}}I_{XY} \tag{13}$$

$$= I_{YX}(I - \Lambda)^{-\frac{1}{2}}R^{-\frac{1}{2}}(I - \Lambda)RR^{-\frac{1}{2}}(I - \Lambda)^{-\frac{1}{2}}I_{XY} \tag{14}$$

$$= I_{YX}I_{XY} \tag{15}$$

$$\Sigma_{\tilde{X}\tilde{Y}} = R^{-\frac{1}{2}}V\Sigma_{XY}\Sigma_Y^{-1}\Sigma_{YX}V^{\mathsf{T}}R^{-\frac{1}{2}}(I - \Lambda)^{-\frac{1}{2}}I_{XY} \tag{16}$$

$$= R^{-\frac{1}{2}}(I - \Lambda)RR^{-\frac{1}{2}}(I - \Lambda)^{-\frac{1}{2}}I_{XY} \tag{17}$$

$$= (I - \Lambda)^{\frac{1}{2}}I_{XY} \tag{18}$$

When the dimension of $X$ is greater than or equal to that of $Y$, $I_{YX}I_{XY} = I$ and we have shown that the canonical transformation is given by two invertible matrices: $K_X = R^{-\frac{1}{2}}V$ and $K_Y = I_{YX}(I - \Lambda)^{-\frac{1}{2}}R^{-\frac{1}{2}}V\Sigma_{XY}\Sigma_Y^{-1}$.

If the dimension of $X$ is smaller than that of $Y$, we can repeat the above analysis but with $X$ and $Y$ in place of one another. That is, we have $K_Y = R_Y^{-\frac{1}{2}}V_Y$ and $K_X = I_{XY}(I - \Lambda_Y)^{-\frac{1}{2}}R_Y^{-\frac{1}{2}}V_Y\Sigma_{YX}\Sigma_X^{-1}$, where $V_Y\Sigma_{Y|X}\Sigma_Y^{-1} = \Lambda_Y V_Y$, $R_Y = V_Y\Sigma_Y V_Y^{\mathsf{T}}$, with both $\Lambda_Y$ and $R_Y$ being diagonal. Again we see that both $K_X$ and $K_Y$ are invertible. $\qquad\square$

Importantly, the invertibility of the canonical transformation preserves information content, $I(T; X) = I(T; \tilde{X})$ and $I(T; Y) = I(T; \tilde{Y})$. As a result, we can reformulate the IB problem for $P_{XY}$ as an equivalent one for $P_{\tilde{X}\tilde{Y}}$. Moreover, since $P_{\tilde{X}\tilde{Y}}$ factorizes, we can obtain an IB solution $Q_{T|X}$ by solving separate IB problems for each bivariate Gaussian variable $(\tilde{X}_i, \tilde{Y}_i)$. That is, $Q_{\tilde{T}|\tilde{X}} = \prod_i Q_{\tilde{T}_i|\tilde{X}_i}$ where $Q_{\tilde{T}_i|\tilde{X}_i}$ is an optimal encoder under the Markov constraint $\tilde{T}_i$—$\tilde{X}_i$—$\tilde{Y}_i$.

### 2.3 Gaussian Information Bottleneck

For Gaussian correlated $X$ and $Y$, Globerson & Tishby (2004) show that an optimal compressed representation of the source $X$ is defined by a linear map with additive Gaussian noise,

$$T = AX + \xi \quad \text{with} \quad \xi \sim \mathcal{N}(0, \Sigma_\xi). \tag{19}$$

Since the mutual information is invariant under an invertible transformation—i.e., $I(T; X) = I(KT; X)$ and $I(T; Y) = I(KT; Y)$ for any invertible $K$—we can always transform $T$ such that the noise covariance matrix $\Sigma_\xi$ becomes an identity matrix $I$, without changing the information content (see Chechik et al. (2005, Appx A)). Without loss of generality, we set $\Sigma_\xi = I$. That is, the encoder becomes a Gaussian channel, parametrized only by the matrix $A$,

$$T \mid X \sim \mathcal{N}(AX, I). \tag{20}$$

This channel completely characterizes the representation $T$; integrating out $X$ from the above equation gives

$$T \sim \mathcal{N}(0, I + A\Sigma_X A^\mathsf{T}) \tag{21}$$

$$T \mid Y \sim \mathcal{N}(A\mu_{X|Y}, I + A\Sigma_{X|Y} A^\mathsf{T}), \tag{22}$$

where $\mu_{X|Y} = \Sigma_{XY}\Sigma_Y^{-1}Y$ and $\Sigma_{X|Y} = \Sigma_X - \Sigma_{XY}\Sigma_Y^{-1}\Sigma_{YX}$.

**Definition 2.4.** *For $\begin{bmatrix} X \\ Y \end{bmatrix} \sim \mathcal{N}(0, \begin{bmatrix} \Sigma_X & \Sigma_{XY} \\ \Sigma_{YX} & \Sigma_Y \end{bmatrix})$, the Gaussian IB loss is given by*

$$\mathrm{GIB}_\beta(A) = I(T; X) - \beta I(T; Y) \quad \text{with} \quad T = AX + \xi, \quad \xi \sim \mathcal{N}(0, I). \tag{23}$$

The following theorem, due to Chechik et al. (2005), provides a solution of the optimization $\min_A \mathrm{GIB}_\beta(A)$.

**Theorem 2.4** [Chechik et al. (2005, Thm 3.1)]. *For $\begin{bmatrix} X \\ Y \end{bmatrix} \sim \mathcal{N}(0, \begin{bmatrix} \Sigma_X & \Sigma_{XY} \\ \Sigma_{YX} & \Sigma_Y \end{bmatrix})$, let $v_i$ and $\lambda_i$ be the left eigenvectors and eigenvalues of $\Sigma_{X|Y}\Sigma_X^{-1}$ (i.e., $v_i^\mathsf{T}\Sigma_{X|Y}\Sigma_X^{-1} = v_i^\mathsf{T}\lambda_i$) and $r_i = v_i^\mathsf{T}\Sigma_X v_i$. Then, a solution of $\min_A \mathrm{GIB}_\beta(A)$ is given by*

$$A = \mathrm{diag}(\alpha_1 r_1^{-\frac{1}{2}}, \alpha_2 r_2^{-\frac{1}{2}}, \cdots) [v_1 \ v_2 \ \cdots]^\mathsf{T} \quad \text{with} \quad \alpha_i = \begin{cases} \sqrt{[\beta(1 - \lambda_i) - 1]/\lambda_i} & \text{if } \beta > 1/(1 - \lambda_i) \\ 0 & \text{otherwise} \end{cases}$$

Our work focuses on extending the above definition and theorem to the case where the mutual information in Eq (23) is replaced by other correlation measures.

**Definition 2.5.** *For $\begin{bmatrix} X \\ Y \end{bmatrix} \sim \mathcal{N}(0, \begin{bmatrix} \Sigma_X & \Sigma_{XY} \\ \Sigma_{YX} & \Sigma_Y \end{bmatrix})$, the generalized Gaussian IB loss reads*

$$\mathrm{GIB}_\beta^{(\Omega)}(A) = \Omega(T; X) - \beta\Omega(T; Y) \quad \text{with} \quad T = AX + \xi, \quad \xi \sim \mathcal{N}(0, I). \tag{24}$$

We note that we can set $\Sigma_\xi = I$ without loss of generality when the mutual correlation measure is invariant under an invertible transformation of a random variable. Both Rényi and Jeffreys generalizations of mutual information in §2.1.1 and §2.1.2 exhibit this property (see Appendix D).

## 3 Rényi Information Bottleneck for Gaussian variables

We now turn to the Rényi generalization of the Gaussian IB problem. Substituting the correlation measure in Eq (24) with Rényi $q$-information yields

$$\mathrm{GIB}_\beta^{(I_q)}(A) = I_q(T; X) - \beta I_q(T; Y) \quad \text{with} \quad T = AX + \xi, \quad \xi \sim \mathcal{N}(0, I). \tag{25}$$

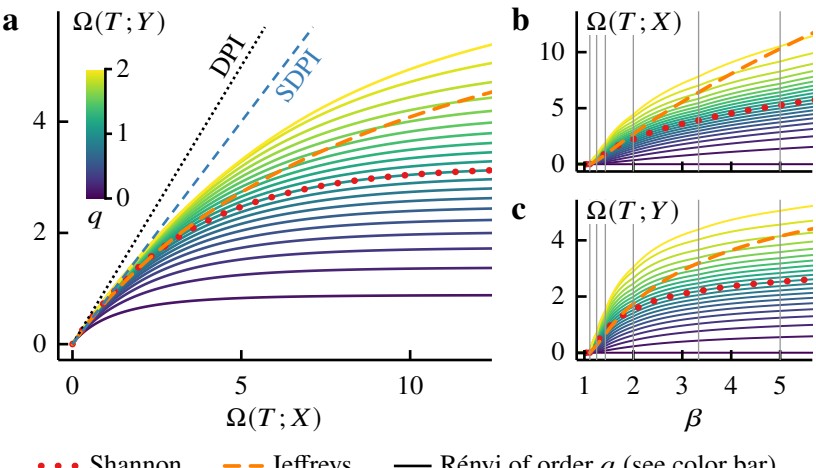

Figure 1: Generalized IB for Gaussian variables yields a solution that depends on the choice of correlation measures. We illustrate IB solutions (Thms 3.1 and 4.1) for a specific instance of normalized regression matrix $\Sigma_{X|Y}\Sigma_X^{-1}$ with eigenvalues $\lambda_i = 0.1, 0.2, 0.3, 0.5, 0.7, 0.8$, for three families of correlation measures (§2.1): Shannon ($\Omega = I$), Jeffreys ($\Omega = J$) and Rényi ($\Omega = I_q$), see legend. For the Rényi case, we show the results for a range of orders $q$ (see color bar). **a** We plot the IB frontier which traces an upper bound on $\Omega(T;Y)$ (how much information a representation $T$ of $X$ can have about $Y$) at any given $\Omega(T;X)$ (the information the representation $T$ has about $X$). We see that this frontier depends on our choice of information measure. The IB frontier is bounded by the data processing inequality (DPI), $\Omega(T;Y) \leq \Omega(T;X)$, and its tight, data-dependent version (SDPI), $\Omega(T;Y) \leq (1 - \lambda_{\min})\Omega(T;X)$, where $\lambda_{\min}$ is the smallest eigenvalue of $\Sigma_{X|Y}\Sigma_X^{-1}$. We emphasize that while Shannon IB is equivalent to Rényi IB with $q = 1$, Jeffreys IB is not a special case of Rényi IB. **b-c** We illustrate $\Omega(T;X)$ and $\Omega(T;Y)$ as a function of the tradeoff parameter $\beta$ [(1)]. Both information terms increase with $\beta$ and vanish below the critical value $\beta_c = 1/(1 - \lambda_{\min})$. The vertical lines mark the critical tradeoff parameters $\beta_c^{(i)} = 1/(1 - \lambda_i)$ [Eq (34)]. We see no nonanalytic behavior in information at these critical points, suggesting that the transitions are continuous in all cases considered.

More generally, the $q$-information terms need not be of the same order but the data processing inequality $I_q(T;X) \geq I_{q'}(T;Y)$ is guaranteed only when $q = q'$.[1]

**Theorem 3.1.** *Let $v_i$ and $\lambda_i$ be the left eigenvectors and eigenvalues of $\Sigma_{X|Y}\Sigma_X^{-1}$ (i.e., $v_i^\mathsf{T}\Sigma_{X|Y}\Sigma_X^{-1} = v_i^\mathsf{T}\lambda_i$), $r_i = v_i^\mathsf{T}\Sigma_X v_i$, and $u_i$ the positive root of*

$$\frac{1}{\beta} = \frac{1 - \lambda_i}{1 + u_i\lambda_i} \frac{1 + \frac{\bar{q}(1+\bar{q})(1-\lambda_i)u_i}{1+(1-\bar{q}^2(1-\lambda_i))u_i}}{1 + \frac{\bar{q}(1+\bar{q})u_i}{1+(1-\bar{q}^2)u_i}}$$

*Then, a solution of the optimization $\min_A \mathrm{GIB}_\beta^{(I_q)}(A)$ reads*

$$A = \mathrm{diag}(\alpha_1 r_1^{-\frac{1}{2}}, \alpha_2 r_2^{-\frac{1}{2}}, \cdots)[v_1 \; v_2 \; \cdots]^\mathsf{T} \quad with \quad \alpha_i = \begin{cases} \sqrt{u_i} & if \; \beta > 1/(1 - \lambda_i) \\ 0 & otherwise \end{cases}$$

*Proof.* First, we consider the scalar case $\begin{bmatrix} X \\ Y \end{bmatrix} \sim \mathcal{N}(0, \begin{bmatrix} 1 & \rho \\ \rho & 1 \end{bmatrix})$. Here the matrix $A$ in Eq (25) reduces to a scalar $\alpha$ and the covariance matrices in Eqs (20-22) become $\Sigma_{T|X} = 1$, $\Sigma_T = 1 + \alpha^2$ and $\Sigma_{T|Y} = 1 + (1 - \rho^2)\alpha^2$. Using the expression for Rényi information in Eq (5), we write down

$$I_q(T;X) = -\frac{1}{2\bar{q}} \ln \frac{(1+\alpha^2)^q}{1+(1-\bar{q}^2)\alpha^2} \tag{26}$$

$$I_q(T;Y) = -\frac{1}{2\bar{q}} \ln \frac{(1+(1-\rho^2)\alpha^2)^{\bar{q}}(1+\alpha^2)^q}{1+(1-\bar{q}^2\rho^2)\alpha^2} \tag{27}$$

---

[1]For $q \neq q'$, $I_q$ and $I_{q'}$ are monotone maps of *different* $f$-divergences, hence the data processing inequality needs not hold.

where $\bar{q} = 1 - q$. Differentiating the above equations with respect to $\alpha$ gives

$$\frac{d}{d\alpha} I_q(T; X) = -\frac{\alpha}{\bar{q}} \left( \frac{q}{1 + \alpha^2} - \frac{1 - \bar{q}^2}{1 + (1 - \bar{q}^2)\alpha^2} \right) \tag{28}$$

$$\frac{d}{d\alpha} I_q(T; Y) = -\frac{\alpha}{\bar{q}} \left( \frac{q}{1 + \alpha^2} + \bar{q}\frac{1 - \rho^2}{1 + \alpha^2(1 - \rho^2)} - \frac{1 - \bar{q}^2\rho^2}{1 + (1 - \bar{q}^2\rho^2)\alpha^2} \right). \tag{29}$$

Equating the derivative of the IB loss to zero, $\frac{d}{d\alpha}[I_q(T; X) - \beta I_q(T; Y)] = 0$, and rearranging the resulting expression yields either $\alpha = 0$ or

$$\frac{1}{\beta} = \frac{dI_q(T; Y)/d\alpha}{dI_q(T; X)/d\alpha} = g_q(\alpha^2, 1 - \rho^2), \tag{30}$$

where we define

$$g_q(u, \lambda) \equiv \frac{1 - \lambda}{1 + u\lambda} \frac{1 + \frac{\bar{q}(1+\bar{q})(1-\lambda)u}{1+(1-\bar{q}^2(1-\lambda))u}}{1 + \frac{\bar{q}(1+\bar{q})u}{1+(1-\bar{q}^2)u}}. \tag{31}$$

For $\lambda \in (0, 1)$, $q \in (0, 1) \cup (1, 2]$ and $u \geq 0$, the function $g_q(u, \lambda)$ is strictly decreasing in $u$, approaching zero as $u \to \infty$ (see Appendix A). As a result, Eq (30) has exactly one positive solution $\alpha^2 > 0$ if $1/\beta < g_q(0, 1 - \rho^2)$. That is, this IB problem admits a nontrivial solution ($\alpha \neq 0$) only when $\beta$ exceeds the critical value

$$\beta_c = \frac{1}{g_q(0, 1 - \rho^2)} = \frac{1}{\rho^2}, \tag{32}$$

which, rather surprisingly, does not depend on $q$.

In the multi-dimensional case, we work in the canonical representation $(\tilde{X}, \tilde{Y}) = (K_X X, K_Y Y)$ (see §2.2). If $\tilde{A}$ is an optimal projection for $P_{\tilde{X}\tilde{Y}}$, then $A = \tilde{A}K_X$ is an optimal linear map for $P_{XY}$ since $I_q(\tilde{A}\tilde{X} + \xi; \tilde{X}) = I_q(\tilde{A}K_X X + \xi; K_X X) = I_q(\tilde{A}K_X X + \xi; X)$, and similarly for $Y$. (The first equality is a direct substitution $\tilde{X} = K_X X$ and the second is due to the fact that $K_X$ is invertible.) The joint distribution $P_{\tilde{X}\tilde{Y}}$ is a product of bivariate Gaussian distributions and thus we can obtain $\tilde{A}$ from the solution of the scalar case. That is, $\tilde{A}\tilde{X} = [\alpha_1 \tilde{X}_1 \ \alpha_2 \tilde{X}_2 \ \cdots]^\mathsf{T}$ or $\tilde{A} = \text{diag}(\alpha_1, \alpha_2, \cdots)$ where $\alpha_i$ is a solution from the scalar case. Recalling that $\sigma_{\tilde{X}_i}^2 = \sigma_{\tilde{Y}_i}^2 = 1$ and $\rho_i = \sqrt{1 - \lambda_i}$ with $\lambda_i$ the eigenvalues of $\Sigma_{X|Y}\Sigma_X^{-1}$, we obtain [see Eq (30)],

$$\alpha_i = 0 \quad \text{if } \beta \leq \beta_c^{(i)} \quad \text{and} \quad \beta^{-1} = g_q(\alpha_i^2, \lambda_i) \quad \text{if } \beta > \beta_c^{(i)} \tag{33}$$

where the critical tradeoff parameter for each dimension is

$$\beta_c^{(i)} = \frac{1}{1 - \lambda_i}. \tag{34}$$

Since $K_X = R^{-\frac{1}{2}}V = \text{diag}(r_1, r_2, \cdots)^{-\frac{1}{2}} [v_1 \ v_2 \ \cdots]^\mathsf{T}$ when the dimension of $X$ is equal to or greater then that of $Y$ (see §2.2), we have

$$A = \tilde{A}K_X = \text{diag}(\alpha_1 r_1^{-\frac{1}{2}}, \alpha_2 r_2^{-\frac{1}{2}}, \cdots) [v_1 \ v_2 \ \cdots]^\mathsf{T}, \tag{35}$$

which is the expression in Thm 3.1. When the dimension of $X$ is less than that of $Y$, we can drop the excess dimensions of $\tilde{Y}$ since they are uncorrelated with $\tilde{X}$ and thus cannot be encoded in any representation of $\tilde{X}$. $\qquad \square$

The optimal projection results in Rényi information

$$I_q(T; X) = -\frac{1}{2\bar{q}} \sum_i^{\beta > \beta_c^{(i)}} \ln \frac{(1 + \alpha_i^2)^q}{1 + (1 - \bar{q}^2)\alpha_i^2} \tag{36}$$

$$I_q(T; Y) = -\frac{1}{2\bar{q}} \sum_i^{\beta > \beta_c^{(i)}} \ln \frac{(1 + \lambda_i \alpha_i^2)^{\bar{q}}(1 + \alpha_i^2)^q}{1 + [1 - \bar{q}^2(1 - \lambda_i)]\alpha_i^2}, \tag{37}$$

where the summations are restricted to the eigenmodes that contribute the IB encoder, i.e., those with $\alpha_i > 0$. We depict an example of the optimal frontiers of Rényi IB in Fig 1.

To conclude our analysis of Rényi IB, we note that the analytical solution of Chechik et al. (2005) is a limiting case of our results. In the limit $q \to 1$, Eq (33) reads

$$\frac{1}{\beta} = g_{q \to 1}(\alpha_i^2, \lambda_i) = \frac{1 - \lambda_i}{1 + \lambda_i \alpha_i^2} \implies \alpha_i^{(q=1)} = \sqrt{\frac{\beta(1 - \lambda_i) - 1}{\lambda_i}}. \tag{38}$$

This solution is identical to that of Chechik et al. (2005, Lemma 4.1), with $\alpha_i / \sqrt{r_i}$ being the weight of each eigenmode.

## 4 Jeffreys Information Bottleneck for Gaussian variables

The technique in the previous section applies also to the IB problems, based on other statistical divergences. In this section, we consider Jeffreys IB for Gaussian variables. Substituting the correlation measure in Eq (24) with Jeffreys information (§2.1.2) yields

$$\mathrm{GIB}_\beta^{(J)}(A) = J(T; X) - \beta J(T; Y) \quad \text{with} \quad T = AX + \xi, \quad \xi \sim \mathcal{N}(0, I). \tag{39}$$

**Theorem 4.1.** *Let $v_i$ and $\lambda_i$ be the left eigenvectors and eigenvalues of $\Sigma_{X|Y} \Sigma_X^{-1}$ (i.e., $v_i^\mathsf{T} \Sigma_{X|Y} \Sigma_X^{-1} = v_i^\mathsf{T} \lambda_i$) and $r_i = v_i^\mathsf{T} \Sigma_X v_i$. Then, a solution of the optimization $\min_A \mathrm{GIB}_\beta^{(J)}(A)$ reads*

$$A = \mathrm{diag}(\alpha_1 r_1^{-\frac{1}{2}}, \alpha_2 r_2^{-\frac{1}{2}}, \cdots) [v_1 \ v_2 \ \cdots]^\mathsf{T} \quad \text{with} \quad \alpha_i = \begin{cases} \sqrt{[\sqrt{\beta(1 - \lambda_i)} - 1]/\lambda_i} & \text{if } \beta > 1/(1 - \lambda_i) \\ 0 & \text{otherwise} \end{cases}$$

*Proof.* We first consider the scalar case $\begin{bmatrix} X \\ Y \end{bmatrix} \sim \mathcal{N}(0, \begin{bmatrix} 1 & \rho \\ \rho & 1 \end{bmatrix})$, for which the projection $A$ is a scalar $\alpha$ and $\Sigma_{T|X} = 1$, $\Sigma_T = 1 + \alpha^2$ and $\Sigma_{T|Y} = 1 + (1 - \rho^2)\alpha^2$. From Eq (9), we have

$$J(T; X) = \frac{1}{2}\alpha^2 \quad \text{and} \quad J(T; Y) = \frac{1}{2}\frac{\rho^2 \alpha^2}{1 + (1 - \rho^2)\alpha^2}. \tag{40}$$

Differentiate the above information with respect to $\alpha$ yields

$$\frac{d}{d\alpha} J(T; X) = \alpha \quad \text{and} \quad \frac{d}{d\alpha} J(T; Y) = \alpha \frac{\rho^2}{(1 + (1 - \rho^2)\alpha^2)^2}. \tag{41}$$

The first order condition, $\frac{d}{d\alpha}[J(T; X) - \beta J(T; Y)] = 0$, results in either $\alpha = 0$ or

$$\alpha^2 = \frac{\sqrt{\beta \rho^2} - 1}{1 - \rho^2}. \tag{42}$$

This equation admits a real $\alpha$ only when $\beta$ exceeds the critical value $\beta_c = 1/\rho^2$, which is identical to that of the Rényi case, see Eq (32).

We can construct an optimal projection $A$ for multi-dimensional $X$ and $Y$, using the canonical representation (§2.2) which factorizes the IB problem into scalar Gaussian IB problems for each eigenmode of $\Sigma_{Y|X} \Sigma_X^{-1}$. Following the same logical steps as in §3, we obtain $A = \mathrm{diag}(\alpha_1 r_1^{-\frac{1}{2}}, \alpha_2 r_2^{-\frac{1}{2}}, \cdots) [v_1 \ v_2 \ \cdots]^\mathsf{T}$, with

$$\alpha_i = 0 \quad \text{if } \beta \leq \beta_c^{(i)} \quad \text{and} \quad \alpha_i = \sqrt{\frac{\sqrt{\beta(1 - \lambda_i)} - 1}{\lambda_i}} \quad \text{if } \beta > (1 - \lambda_i)^{-1}. \tag{43}$$

We note that the Jeffreys IB method for Gaussian variables admits the exact same set of critical tradeoff parameters, $\beta_c^{(i)} = (1 - \lambda_i)^{-1}$, as the Rényi case, see Eq (34). □

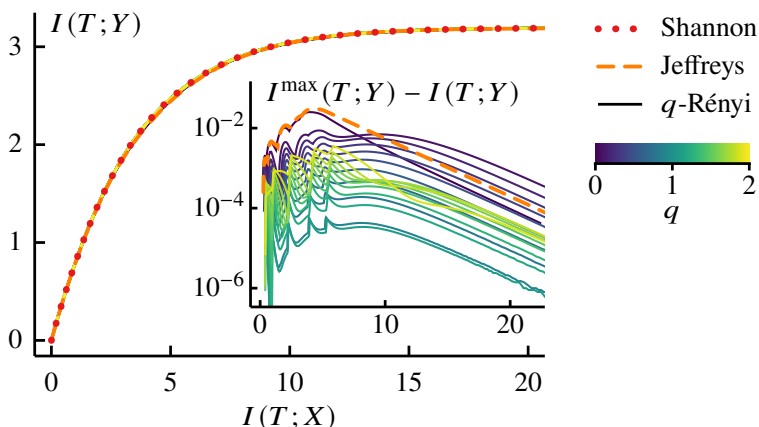

Figure 2: Rényi and Jeffreys IB problems for Gaussian variables admit solutions that are close to Shannon IB optimal. Plotted on the Shannon information plane, the solutions to Shannon (dotted), Jeffreys (dashed) and Rényi (solid) IB problems are nearly indistinguishable. For the Rényi case, we depict the results for a range of Rényi orders $q$ (see color bar). *Inset:* We depict the gap between the maximum achievable and encoded relevant Shannon informations, $I^{\max}(T;Y)$ and $I(T;Y)$ respectively, as a function of the extracted Shannon information $I(T;X)$. This gap vanishes in the low and high-information limits, $I(T;X) \to 0$ and $I(T;X) \to \infty$. The satellite peaks result from the fact that the solutions to Shannon, Jeffreys and Rényi IB problems go through structural transitions at different values of $I(T;X)$ even though these transitions occur at the same set of critical tradeoff parameters. Here the eigenvalues of $\Sigma_{X|Y}\Sigma_X^{-1}$ are $\lambda_i = 0.1, 0.2, 0.3, 0.5, 0.7, 0.8$.

Finally, we write down Jeffreys information using the optimal projection,

$$J(T;X) = \frac{1}{2} \sum_i^{\beta > \beta_c^{(i)}} \frac{\sqrt{\beta(1-\lambda_i)} - 1}{\lambda_i} \tag{44}$$

$$J(T;Y) = \frac{1}{2} \sum_i^{\beta > \beta_c^{(i)}} \frac{1-\lambda_i}{\lambda_i} \frac{\sqrt{\beta(1-\lambda_i)} - 1}{\sqrt{\beta(1-\lambda_i)}}. \tag{45}$$

where the summations are limited to the modes that contribute to the encoder, i.e., those with $\beta_c^{(i)} < \beta$. In Fig 1, we depict an example of the Jeffreys IB optimal frontier, computed from the above equations. We emphasize that while Shannon information is equivalent to Rényi information with $q = 1$, Jeffreys information is not a special case of Rényi information.

## 5  Discussion & Conclusion

In Fig 2, we depict the optimal Gaussian encoders for the original, Rényi and Jeffreys IB problems on the Shannon information plane. We see that these solutions are very close to the optimal frontier, characterized by the Shannon IB solutions. This result suggests that formulating and solving an IB problem, defined with alternative correlation measures other than Shannon information, could offer a strategy for obtaining an approximate solution to the original IB problem. To better illustrate the differences between the solutions to the original, Rényi and Jeffreys IB problems, the inset shows how much less relevant Shannon information the optimal linear representations of Rényi and Jeffreys IB encode, compared to the Shannon IB optimal representation. We see that the differences are maximum at intermediate information and vanish in the low and high-information limits. In addition, the Shannon information gaps exhibit satellite peaks, resulting from structural the transition of the IB solutions. We note that although these transitions occur at the same critical tradeoff parameters $\beta_c^{(i)} = 1/(1-\lambda_i)$, they generally correspond to different values of extracted Shannon information.

To sum up, we consider generalized IB problems in which the mutual information is replaced by mutual dependence measures, based on Rényi and Jeffreys divergences. We obtain exact analytical IB optimal Gaussian encoders for the case of Gaussian correlated random variables, generalizing the results of Chechik et al. (2005). We show that the fundamental IB tradeoff between relevance and compression holds also for correlation measures other than Shannon information. Our analyses reveal structural transitions in the optimal representations, similar to that in the original IB method (Chechik et al., 2005). Interestingly the critical tradeoff parameters are the same for original, Rényi and Jeffreys IB problems, even though the solutions are distinct.

We anticipate that our work will find application in physics of correlated components which relies on Rényi-generalization of entropy and information to quantify entanglement. In addition, our characterization of Jeffreys IB could have implications for understanding the generalization properties of Gibbs learning algorithms of which the generalization gap is proportional to Jeffreys information between fitted models and training data. Finally we note that the conditional IB problem, in which the compression term $I(T; X)$ is replaced by $I(T; X \,|\, Y)$, becomes non-trivial for generalized information measures since the chain rule does not hold for Rényi and Jeffreys information—that is, given the Markov constraint $T$–$X$–$Y$, we have $I(T; X \,|\, Y) = I(T; X) - I(T; Y)$ for Shannon information, but in general, $\Omega(T; X \,|\, Y) \neq \Omega(T; X) - \Omega(T; Y)$. The logical steps in our analyses are readily generalizable to conditional IB problems.

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

# A  Supplementary figure

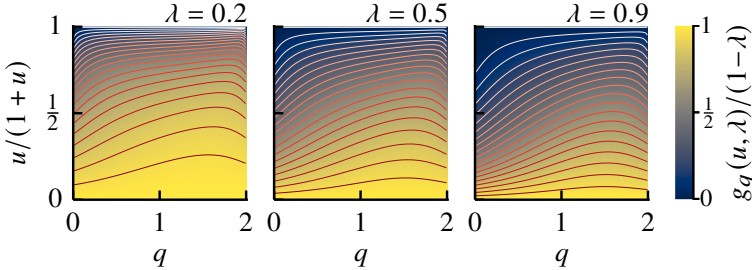

Figure A1: The function $g_q(u, \lambda)$ [Eq (31)] decreases with $u$ from $1 - \lambda$ at $u = 0$ and approaches zero as $u \to \infty$. As a result, the equation $\beta^{-1} = g_q(u, \lambda)$ always has a unique positive solution when $\beta > 1/(1 - \lambda)$. We consider only $0 \leq q \leq 2$ since Rényi information for Gaussian variables can diverge for $q > 2$ [see Eq (5)].

# B  Rényi information for Gaussian variables

In this appendix, we derive Rényi mutual information for Gaussian correlated variables. Using the definition from Eqs (3-4), we write down Rényi mutual information for continuous random variables,

$$I_q(X;Y) = \frac{1}{q-1} \ln \int dx dy \, p_X(x) p_Y(y) \left( \frac{p_{XY}(x,y)}{p_X(x) p_Y(y)} \right)^q. \tag{46}$$

where $p_X$, $p_Y$ and $p_{XY}$ denote the probability density functions of $X$, $Y$ and $(X, Y)$, respectively. We consider Gaussian correlated random variables

$$\begin{bmatrix} X \\ Y \end{bmatrix} \sim \mathcal{N}(\mu, \Sigma) \ \text{ with } \ \mu = \begin{bmatrix} \mu_X \\ \mu_Y \end{bmatrix} \ \text{ and } \ \Sigma = \begin{bmatrix} \Sigma_X & \Sigma_{XY} \\ \Sigma_{YX} & \Sigma_Y \end{bmatrix}. \tag{47}$$

In this case, the joint probability density is given by

$$p_{XY}(x,y) = \frac{\exp\left\{ -\frac{1}{2} \left( \begin{bmatrix} x \\ y \end{bmatrix} - \mu \right)^\mathsf{T} \Sigma^{-1} \left( \begin{bmatrix} x \\ y \end{bmatrix} - \mu \right) \right\}}{|2\pi\Sigma|^{1/2}} \tag{48}$$

The product of the marginal distributions is equal to a joint distribution but with $\Sigma_{XY}$ and $\Sigma_{YX}$ set to zero, i.e.,

$$p_X(x) p_Y(y) = \frac{\exp\left\{ -\frac{1}{2} \left( \begin{bmatrix} x \\ y \end{bmatrix} - \mu \right)^\mathsf{T} \bar{\Sigma}^{-1} \left( \begin{bmatrix} x \\ y \end{bmatrix} - \mu \right) \right\}}{|2\pi\bar{\Sigma}|^{1/2}} \tag{49}$$

where $\bar{\Sigma} = \begin{bmatrix} \Sigma_X & \cdot \\ \cdot & \Sigma_Y \end{bmatrix}$. Substituting the above densities into Eq (46) and performing the resulting Gaussian integration over $x$ and $y$ gives

$$I_q(X;Y) = \frac{1}{q-1} \ln \frac{|q\Sigma^{-1} + (1-q)\bar{\Sigma}^{-1}|^{-1/2}}{|\Sigma|^{q/2} |\bar{\Sigma}|^{(1-q)/2}}. \tag{50}$$

The determinants of the covariance matrices are given by

$$|\Sigma| = |\Sigma_Y| \times |\Sigma_{X|Y}| \quad \text{and} \quad |\bar{\Sigma}| = |\Sigma_Y| \times |\Sigma_X|, \tag{51}$$

where $\Sigma_{X|Y} = \Sigma_X - \Sigma_{XY} \Sigma_Y^{-1} \Sigma_{YX}$ and we use Schur's formula

$$\begin{vmatrix} A & B \\ C & D \end{vmatrix} = |D| \times |A - BD^{-1}C|. \tag{52}$$

We now consider the numerator in Eq (50),

$$
\begin{aligned}
\left| q\Sigma^{-1} + (1-q)\bar{\Sigma}^{-1} \right| &= \left| \Sigma^{-1} \left( q\bar{\Sigma} + (1-q)\Sigma \right) \bar{\Sigma}^{-1} \right| \\
&= \frac{1}{|\Sigma| \times |\bar{\Sigma}|} \left| \begin{matrix} \Sigma_X & (1-q)\Sigma_{XY} \\ (1-q)\Sigma_{YX} & \Sigma_Y \end{matrix} \right| \\
&= \frac{\left| I - (1-q)^2 (I - \Sigma_{X|Y}\Sigma_X^{-1}) \right|}{|\Sigma_Y| \times |\Sigma_{X|Y}|},
\end{aligned}
\tag{53}
$$

where the last equality follows from Eqs (51-52). Finally we write down the Rényi information for Gaussian variables

$$
I_q(X;Y) = \frac{1/2}{q-1} \ln \frac{|\Sigma_{X|Y}\Sigma_X^{-1}|^{1-q}}{\left| I - (1-q)^2(I - \Sigma_{X|Y}\Sigma_X^{-1}) \right|}.
\tag{54}
$$

This expression is identical to Eq (5) (with $\bar{q} = 1 - q$).

## B.1 Alternative definition of Rényi information

An alternative generalization of Shannon mutual information is based on its entropy representation,

$$
I(X;Y) = S(X) + S(Y) - S(X,Y),
\tag{55}
$$

where the terms on the *rhs* denote the entropy of the random variables $X$, $Y$ and $(X,Y)$, respectively. Replacing the entropy in the above equation by Rényi entropy,

$$
S_q(X) = \frac{1}{1-q} \times \left\{ \begin{matrix} \ln \sum_x p_X(x)^q & \text{for discrete } X \\ \ln \int dx\, p_X(x)^q & \text{for continuous } X \end{matrix} \right. ,
\tag{56}
$$

yields a definition of Rényi mutual information,

$$
\tilde{I}_q(X;Y) = S_q(X) + S_q(Y) - S_q(X,Y),
\tag{57}
$$

where $q \in (0,1) \cup (1, \infty)$.

This definition is commonly used in physics literature (see, e.g., Brydges et al. (2019, Eq 4)), but it gives Rényi information that is equal to Shannon information regardless of $q$ for Gaussian correlated $X$ and $Y$. To see this, we recall the probability density of a Gaussian variable $X \sim \mathcal{N}(\mu_X, \Sigma_X)$,

$$
p_X(x) = |2\pi\Sigma_X|^{-1/2} e^{-\frac{1}{2}(x-\mu_X)^{\mathsf{T}}\Sigma_X^{-1}(x-\mu_X)}.
$$

Substituting the above density into Eq (56) and using the multidimensional Gaussian integral to integrate out $x$ leads to

$$
S_q(X) = S(X) - \frac{n_X}{2}\left(1 + \frac{\ln q}{1-q}\right)
\tag{58}
$$

where $n_X$ is the dimensions of $X$ and $S(X) = \frac{1}{2}\ln|2\pi e\Sigma_X|$ is the differential entropy of a Gaussian vector. Similarly, for $\begin{bmatrix} X \\ Y \end{bmatrix} \sim \mathcal{N}(\begin{bmatrix} \mu_X \\ \mu_Y \end{bmatrix}, \begin{bmatrix} \Sigma_X & \Sigma_{XY} \\ \Sigma_{YX} & \Sigma_Y \end{bmatrix})$, we have

$$
S_q(Y) = S(Y) - \frac{n_Y}{2}\left(1 + \frac{\ln q}{1-q}\right)
\tag{59}
$$

$$
S_q(X,Y) = S(X,Y) - \frac{n_X + n_Y}{2}\left(1 + \frac{\ln q}{1-q}\right),
\tag{60}
$$

where $n_Y$ denotes the dimensions of $Y$. Substituting the above expressions into Eq (57) yields

$$
\tilde{I}_q(X;Y) = S(X) + S(Y) - S(X,Y) = I(X;Y),
\tag{61}
$$

where the last equation is due to Eq (55). We see that for Gaussian correlated variables, the Rényi information defined in Eq (57) is identical to Shannon information for $q \in (0,1) \cup (1,\infty)$.

## C   Jeffreys information for Gaussian variables

The Jeffreys information is defined via

$$J(X;Y) \equiv D_{\mathrm{J}}(P_{XY} \parallel P_X \otimes P_Y), \tag{62}$$

where $D_{\mathrm{J}}$ is Jeffreys divergence (Jeffreys, 1946),

$$D_{\mathrm{J}}(P \parallel Q) = \frac{1}{2}[D_{\mathrm{KL}}(P \parallel Q) + D_{\mathrm{KL}}(Q \parallel P)]. \tag{63}$$

For Gaussian correlated $X$ and $Y$, the Jeffreys information follows immediately from the KL divergence between two multivariate Gaussian distributions

$$D_{\mathrm{KL}}(\mathcal{N}(\mu_0, \Sigma_0) \parallel \mathcal{N}(\mu_1, \Sigma_1)) = \frac{1}{2}\left( \mathrm{tr}(\Sigma_1^{-1}\Sigma_0 - I) + (\mu_1 - \mu_0)^{\mathsf{T}}\Sigma_1^{-1}(\mu_1 - \mu_0) + \ln\frac{|\Sigma_1|}{|\Sigma_0|} \right). \tag{64}$$

For $X$ and $Y$ described by Eq (47), we have $P_{XY} = \mathcal{N}(\mu, \Sigma)$ and $P_X \otimes P_Y = \mathcal{N}(\mu, \bar{\Sigma})$, where $\Sigma = \begin{bmatrix} \Sigma_X & \Sigma_{XY} \\ \Sigma_{YX} & \Sigma_Y \end{bmatrix}$ and $\bar{\Sigma} = \begin{bmatrix} \Sigma_X & \cdot \\ \cdot & \Sigma_Y \end{bmatrix}$. As a result, we have

$$D_{\mathrm{KL}}(P_{XY} \parallel P_X \otimes P_Y) = \frac{1}{2}\left( \mathrm{tr}(\bar{\Sigma}^{-1}\Sigma - I) + \ln\frac{|\bar{\Sigma}|}{|\Sigma|} \right) \tag{65}$$

$$D_{\mathrm{KL}}(P_X \otimes P_Y \parallel P_{XY}) = \frac{1}{2}\left( \mathrm{tr}(\Sigma^{-1}\bar{\Sigma} - I) + \ln\frac{|\Sigma|}{|\bar{\Sigma}|} \right). \tag{66}$$

We see that the logarithmic term drops out upon symmetrization [Eq (63)]. Substituting $\bar{\Sigma}^{-1} = \begin{bmatrix} \Sigma_X^{-1} & \cdot \\ \cdot & \Sigma_Y^{-1} \end{bmatrix}$ and the determinant formula in Eq (51) into Eq (65) gives

$$D_{\mathrm{KL}}(P_{XY} \parallel P_X \otimes P_Y) = \frac{1}{2}\ln\frac{|\bar{\Sigma}|}{|\Sigma|} = -\frac{1}{2}\ln|\Sigma_{X|Y}\Sigma_X^{-1}| \tag{67}$$

which is the usual mutual information, as expected. To compute the trace in Eq (66), we write down the inverse of the covariance matrix,

$$\Sigma^{-1} = \begin{pmatrix} \Sigma_{X|Y}^{-1} & -\Sigma_{X|Y}^{-1}\Sigma_{XY}\Sigma_Y^{-1} \\ -\Sigma_Y^{-1}\Sigma_{YX}\Sigma_{X|Y}^{-1} & \Sigma_{Y|X}^{-1} \end{pmatrix}. \tag{68}$$

Therefore we have

$$\begin{aligned} \mathrm{tr}(\Sigma^{-1}\bar{\Sigma} - I) &= \mathrm{tr}(\Sigma^{-1}(\bar{\Sigma} - \Sigma)) \\ &= \mathrm{tr}\left( \begin{bmatrix} \Sigma_{X|Y}^{-1} & -\Sigma_{X|Y}^{-1}\Sigma_{XY}\Sigma_Y^{-1} \\ -\Sigma_Y^{-1}\Sigma_{YX}\Sigma_{X|Y}^{-1} & \Sigma_{Y|X}^{-1} \end{bmatrix} \begin{bmatrix} \cdot & -\Sigma_{XY} \\ -\Sigma_{YX} & \cdot \end{bmatrix} \right) \\ &= \mathrm{tr}(\Sigma_{X|Y}^{-1}\Sigma_{XY}\Sigma_Y^{-1}\Sigma_{YX}) + \mathrm{tr}(\Sigma_Y^{-1}\Sigma_{YX}\Sigma_{X|Y}^{-1}\Sigma_{XY}) \\ &= 2\,\mathrm{tr}(\Sigma_{XY}\Sigma_Y^{-1}\Sigma_{YX}\Sigma_{X|Y}^{-1}) \\ &= 2\,\mathrm{tr}(\Sigma_X\Sigma_{X|Y}^{-1} - I), \end{aligned} \tag{69}$$

where the last equality follows from the identity $\Sigma_{X|Y} = \Sigma_X - \Sigma_{XY}\Sigma_Y^{-1}\Sigma_{YX}$. Substituting the above result into Eq (66) yields

$$D_{\mathrm{KL}}(P_X \otimes P_Y \parallel P_{XY}) = \mathrm{tr}(\Sigma_X\Sigma_{X|Y}^{-1} - I) + \frac{1}{2}\ln|\Sigma_{X|Y}\Sigma_X^{-1}|. \tag{70}$$

Finally eliminating the logarithmic term with Eq (67) leads to

$$\begin{aligned} J(X;Y) &= \frac{1}{2}[D_{\mathrm{KL}}(P_X \otimes P_Y \parallel P_{XY}) + D_{\mathrm{KL}}(P_{XY} \parallel P_X \otimes P_Y)] \\ &= \frac{1}{2}\mathrm{tr}\left( \Sigma_X\Sigma_{X|Y}^{-1} - I \right). \end{aligned} \tag{71}$$

# D    Correlation measures under invertible transformation

**Lemma D.1** [Liese & Vajda (2006, Thm 14); Qiao & Minematsu (2010, Thm 1)]. *Let $x \in \mathcal{X}$, and $p_X$ and $q_X$ be probability densities. The integral $\int_{\mathcal{X}} dx \, q_X(x) f(p_X(x)/q_X(x))$ is invariant under an invertible and differentiable transformation $T : \mathcal{X} \to \tilde{\mathcal{X}}$.*

*Proof.* Let $\tilde{x} = T(x)$. Then, $dx = |J_T(\tilde{x})| d\tilde{x}$, where $|J_T(\tilde{x})|$ denotes the Jacobian determinant. Let $\rho_X$ be a probability density function, the conservation of probability, $\rho(x) dx = \rho_{\tilde{X}}(x) d\tilde{x}$, implies $\rho_X(x) = \rho_{\tilde{X}}(\tilde{x}) |J_T(\tilde{x})|^{-1}$, where $\rho_{\tilde{X}}$ is the corresponding probability density function of $\tilde{x}$. As a result,

$$\int_{\mathcal{X}} dx \, q_X(x) f\left(\frac{p_X(x)}{q_X(x)}\right) = \int_{\tilde{\mathcal{X}}} d\tilde{x} \, q_{\tilde{X}}(\tilde{x}) f\left(\frac{p_{\tilde{X}}(\tilde{x})|J_T(\tilde{x})|^{-1}}{q_{\tilde{X}}(\tilde{x})|J_T(\tilde{x})|^{-1}}\right) = \int_{\tilde{\mathcal{X}}} d\tilde{x} \, q_{\tilde{X}}(\tilde{x}) f\left(\frac{p_{\tilde{X}}(\tilde{x})}{q_{\tilde{X}}(\tilde{x})}\right).$$

$\square$

Since $f$-divergence and Rényi divergence depend on the probability densities only through the integral $\int dx \, q_X(x) f(p_X(x)/q_X(x))$, they are invariant under an invertible and differentiable transformation of the random variable $X$. Consequently, the correlation measures, defined with these divergences, are also invariant under an invertible and differentiable map.

This invariance property means that we can manipulate the covariance matrix of a random vector without changing the information content. For random vectors, $A$ and $B$, a random variable $C$ and an invertible matrix $K$, we have

$$\Omega(A + B; C) = \Omega(KA + KB; C) = \Omega(\tilde{A} + \tilde{B}; C),$$

where $\tilde{A} = KA$ and $\tilde{B} = KB$. Then, the covariance matrix of $\tilde{B}$ is $\Sigma_{\tilde{B}} = K\Sigma_B K^{\mathsf{T}}$. If we choose $K = \Sigma_B^{-\frac{1}{2}}$, it follows that $\Sigma_{\tilde{B}} = I$. That is, we can transform $A$ and $B$ into new random variables $\tilde{A}$ and $\tilde{B}$ such that the information $\Omega(A + B; C)$ is unchanged and the covariance of $\tilde{B}$ is an identity matrix.

