# OpenReview forum: "Generalized Information Bottleneck for Gaussian Variables"
_TMLR — Rejected by TMLR_

### Review · Reviewer_Sa5s · 2023-04-14

**Summary Of Contributions:**

The authors extend the information bottleneck (IB) framework to encompass correlation measures based on Rényi and Jeffreys divergences. This leads to two separate generalisations of the original IB problem. The authors derive an exact analytical solution for both generalisations in the particular case of Gaussian correlated variables. They identify structural transitions that are similar to those previously observed in the original IB case.

**Audience:**

Yes

**Broader Impact Concerns:**

I do not see any concerns on the ethical implications of the work.

**Claims And Evidence:**

No

**Requested Changes:**

The following proposed adjustments are critical to me:

1. Add a background section according to the suggestions made for **2 Divergence-based Correlation measure** and **3.1 Gaussian variables** above.

2. Address the main points described in **4 Rényi Information Bottleneck for Gaussian variables** (i.e. 3,4,6,8,9,10,12,13) and reflect those changes in **5 Jeffreys Information Bottleneck for Gaussian variables** as well.

Other adjustments that would strengthen the work are the remaining adjustments listed in **Strengths And Weaknesses** above.


**Strengths And Weaknesses:**

The idea of going beyond the original IB case is interesting and could be relevant to the community. Nevertheless, I think the clarity of the manuscript should be improved. In particular:

1. The manuscript should be more self-contained. I was expecting to see a background section which recalls the notation and the analytical solution to the original IB problem in the Gaussian correlated random variables case. It should also be clearer why the case of Gaussian correlated random variables is important in large scale scenarios.

2. The novel results derived in the manuscript should be written more rigorously. For example, I think that some major clarifications are needed in Section 4.

Please find below a detailed review regarding the aspects that I think could be improved.

### Abstract

*“Here we consider a generalized IB problem, in which the mutual information in the original IB method is replaced by correlation measures based on Rényi and Jeffreys divergences.”*

Consider clarifying the fact that two separate generalisations of the original IB problem are being considered, one based on the Rényi divergence (Rényi IB problem) and one based on the Jeffreys divergence (Jeffreys IB problem).

### 1 Information Bottleneck

1. Consider renaming this section Introduction.

2. I think the description of the contributions should be made clearer in the last two paragraphs. For example,

* Has anyone besides the authors attempted to replace the Shannon MI by some form of mutual independence measure?

* Why is the case of Gaussian correlated random variables important in large scale applications, besides the fact that it leads to an analytical solution for the original IB problem?

* Could the authors clarify what they mean by well-defined when they write that *“the IB optimization, Eq (1), remains well-defined when the information terms are replaced by appropriate mutual dependence measures”* (i.e. is it tied to the data processing inequality, does it need to be proven?)?

* In the last paragraph, consider insisting on the fact that the Rényi and Jeffreys IB problems are solved separately when $X$ and $Y$ are Gaussian correlated random variables.

### 2 Divergence-based Correlation measure

1. For the sake of completeness and to increase the clarity of the paper, I think that the authors should have a separate background section. A first subsection would be devoted to providing the notation and recalling the main result obtained in the original IB problem for Gaussian Information Bottleneck (Theorem 3.1 of Chechik et al., 2005). A second subsection would introduce the concept of divergence-based correlation measure (and even the resulting general IB problem?), with the Rényi $q$–information and Jeffreys information as examples of divergence-based correlation measures.

2. I would have expected to see the notation $\Omega(X; Y)$ used to define the general resulting IB problem and to see written somewhere an explanation as to when/why this general IB problem is “well-defined”.

### 3.1 Gaussian variables

1. Is the derivation written in (5) novel? Consider moving 3.1 within Section 4 and writing the result given in (5) in a Lemma. Also indicate that $|x|$ denotes the determinant of $x$.

2. Could the authors provide references regarding alternative definitions of the Rényi mutual information such as the Rényi (differential) entropy of order $q$? What is $S_q(X, Y )$ for the Rényi (differential) entropy of order $q$? Can the authors add a proof in appendix for the fact that the Rényi (differential) entropy of order $q$ reduces to Shannon information in the Gaussian correlated case?

3. (39) should be in the main text instead of in Appendix B.

### 4 Rényi Information Bottleneck for Gaussian variables

1. *“Replacing the mutual information in the original IB objective [Eq (1)] with $q$-information yields”* Please add “Rényi” before “$q$-information”.

2. Can the authors explain why/provide a reference for *“the data processing inequality $I_q(T ; X) ≥ I_{q′}(T ; Y )$ is guaranteed only when $q = q′$”*?

3. Please state the main result in a theorem environment (e.g. following Theorem 3.1 of Chechik et al., 2005) with the required assumptions and please also write the proof in a separate proof environment or subsection.

4. *“Since Rényi divergences are invariant under an invertible transformation of random variables [see Eq (4)], we can transform $T$ such that $\Sigma_\xi$ becomes the identity matrix without changing the information content.”* Can the authors provide the proof of this?

5. “We note that $V \Sigma_X^{1/2}$ is orthogonal and thus $V \Sigma_X V^T$ is a diagonal matrix (Chechik et al., 2005, Lemma B.1)”. Do you mean that by (Chechik et al., 2005, Lemma B.1) $V \Sigma_X V^T$ is a diagonal matrix, which is proved in the proof of (Chechik et al., 2005, Lemma B.1) by showing that $V \Sigma_X^{1/2}$ is orthogonal?

6. Eq (12) in Chechik et al. (2005) justifies writing $A$ under the form $A =WV$. Do the authors have a similar result for the Rényi IB problem that I may have missed?

7. Eq (19) is confusing. Please use the notation $M^{-1}$ for the inverse of a matrix $M$ instead of a ratio $1/M$.

8. *“We also assume that $R$ and $W$ are invertible”*. Please add this assumption when stating your theorem. What happens in the case where $R$ and $W$ are not assumed to be invertible? Can you adapt the reasoning from Lemma D.1 of Chechik et al. (2005), in which they distinguish between the case where $W$ is full rank ($k = n_x$) and when it is not ($k < n_x$).

9. Can the authors justify why they seek a solution of the form (20)? Or is it an assumption?

10. The substitution in (19) leading to (21) is not straightforward for me. It would help to have intermediate results showing how (19) simplifies to (21).

11. *“ansatz”* seems to be a typo.

12. I have trouble reconciling the proof made when $R$ and $W$ are invertible with the following paragraph *“Although the above calculation does not uniquely determine the mixing matrix $W$…where the summations are restricted to the eigenmodes that contribute the IB encoder, i.e., those with $u_i > 0$”*. Under the assumption that $R$ and $W$ are invertible, shouldn’t we have that all $u_i >0$?

13. *“To complete our analysis of Rényi IB, we note that the analytical solution of Chechik et al. (2005) is a limiting case of our results.”* Following my previous point, is the analysis of Rényi IB done by the authors a limiting case of Chechik et al. (2005) **in the particular setting where $R$ and $W$ are assumed to be invertible**?

### 5 Jeffreys Information Bottleneck for Gaussian variables

1. I believe the comments I made earlier apply to section 5 as well. For example, **3.1 Gaussian variables, 1.** for (30), **4 Rényi Information Bottleneck for Gaussian variables, 3,4, 6, 7** [for (32) and (33)], **8...**.

2. In the equation after (33), consider changing $A$ to something else, as $A$ is a reserved notation.

### Appendix B

Consider referring to (44) as the Schur complement formula.

---

> ### Author Response · Authors · 2023-05-22
> **Author Response (1/3)**
>
> We are pleased that you find our work relevant and interesting. We appreciate your comments and suggestions which have helped us improve the presentation of our paper. We have addressed all comments point-by-point, making edits as suggested. We hope our response is satisfactory. Please, let us know if we can further clarify any points.
>
> &nbsp;
>
> > ### Abstract
> > _“Here we consider a generalized IB problem, in which the mutual information in the original IB method is replaced by correlation measures based on Rényi and Jeffreys divergences.”_
> >
> > Consider clarifying the fact that two separate generalisations of the original IB problem are being considered, one based on the Rényi divergence (Rényi IB problem) and one based on the Jeffreys divergence (Jeffreys IB problem).
>
> We have clarified this point in our revised abstract.
>
> &nbsp;
>
> >###  1 Information Bottleneck
> > 1. Consider renaming this section Introduction.
>
> We renamed this section, as suggested.
>
> > 2. I think the description of the contributions should be made clearer in the last two paragraphs. For example,
> > - Has anyone besides the authors attempted to replace the Shannon MI by some form of mutual independence measure?
>
> We have added two references (Harremöes & Tishby _ISIT_ 2007 and Hsu et al _ISIT_ 2018) which considered other correlation measures in the IB method.
>
> > - Why is the case of Gaussian correlated random variables important in large scale applications, besides the fact that it leads to an analytical solution for the original IB problem?
>
> We would like to point out that it is the existence of an analytical solution that makes the Gaussian case relevant in applications. In many interesting settings, it is possible to control the data. The IB solution for Gaussian correlated data allows for a principled and exact investigation of the optimality and adaptability of learning systems. See, _eg,_ Palmer et al [*Predictive information in a sensory population*](https://www.pnas.org/doi/10.1073/pnas.1506855112) PNAS 2015. We clarify this point in our revised manuscript.
>
> >  - Could the authors clarify what they mean by well-defined when they write that _“the IB optimization, Eq (1), remains well-defined when the information terms are replaced by appropriate mutual dependence measures”_ (i.e. is it tied to the data processing inequality, does it need to be proven?)?
>
> We have removed this imprecise statement.
>
> > - In the last paragraph, consider insisting on the fact that the Rényi and Jeffreys IB problems are solved separately when $X$ and $Y$ are Gaussian correlated random variables.
>
> We have made this point clear in our updated manuscript.
>
> &nbsp;
>
> > ### 2 Divergence-based Correlation measure
> > 1. For the sake of completeness and to increase the clarity of the paper, I think that the authors should have a separate background section. A first subsection would be devoted to providing the notation and recalling the main result obtained in the original IB problem for Gaussian Information Bottleneck (Theorem 3.1 of Chechik et al., 2005). A second subsection would introduce the concept of divergence-based correlation measure (and even the resulting general IB problem?), with the Rényi–information and Jeffreys information as examples of divergence-based correlation measures.
>
> We added a new background section in our updated paper. This section has three subsections: **1 Divergence-based correlation measures** provides the definitions of Rényi and Jeffreys information as well as their expressions for the Gaussian variables; **2 Canonical representation for Gaussian variables** recalls the properties of Gaussian variables that are key to the Gaussian IB method; **3 Gaussian IB** recaps the results of Chechik et al (2005) and defines the problem we address in our work.
>
> > 2. I would have expected to see the notation $\Omega(X;Y)$ used to define the general resulting IB problem and to see written somewhere an explanation as to when/why this general IB problem is _“well-defined”_.
>
> We now use this notation in definitions 2.1 and 2.5 to define the generalized IB problem and its Gaussian case, respectively. We have removed the imprecise claim that generalized IB is _'well-defined'_.

---

> ### Author Response · Authors · 2023-05-22
> **Author Response (2/3)**
>
> >### 3.1 Gaussian variables
> > 1. Is the derivation written in (5) novel? Consider moving 3.1 within Section 4 and writing the result given in (5) in a Lemma. Also indicate that $|x|$ denotes the determinant of $x$.
>
> We state this result as Lemma 2.1, which includes the definition of the notation for determinant. Since this result follows from a straightforward algebraic manipulation (Appx B), we do not claim novelty here. For the Jeffreys case, the corresponding result and proof are in Lemma 2.2 and Appx C, respectively.
>
> > 2. Could the authors provide references regarding alternative definitions of the Rényi mutual information such as the Rényi (differential) entropy of order $q$? What is $S_q(X,Y)$ for the Rényi (differential) entropy of order $q$? Can the authors add a proof in appendix for the fact that the Rényi (differential) entropy of order $q$ reduces to Shannon information in the Gaussian correlated case?
>
> This definition of Rényi information appears in, _eg_, Eq (4) in Brydges et al [_Probing Rényi entanglement entropy via randomized measurements_](https://www.science.org/doi/10.1126/science.aau4963) Science 2019. In our revised manuscript, we refer to this reference where we discuss this variant of Rényi information.
>
> The new Appx B.1 provides a more detailed discussion of this form of Rényi information and a proof that it is identical to Shannon information for Gaussian variables.
>
> > 3. (39) should be in the main text instead of in Appendix B.
>
> We have included this equation in the main text in Sec 2.2, where we recap the properties of Gaussian correlated variables.
>
> >### 4 Rényi Information Bottleneck for Gaussian variables
> > 1. “_Replacing the mutual information in the original IB objective [Eq (1)] with q-information yields_” Please add “_Rényi_” before “_q-information_”.
>
> In the updated paper, we have made sure to add *Rényi* everywhere we refer to *q-information*.
>
> > 2. Can the authors explain why/provide a reference for “_the data processing inequality $I_q(T;X) \ge I_q'(T;Y)$ is guaranteed only when $q=q'$_”?
>
> The data processing inequality (DPI) is guaranteed when the information measures are defined with the same *f*-divergence. This condition, and thus the DPI, no longer holds when *q*≠*q'*. We have added this explanation as a footnote.
>
> > 3. Please state the main result in a theorem environment (e.g. following Theorem 3.1 of Chechik et al., 2005) with the required assumptions and please also write the proof in a separate proof environment or subsection.
>
> In the updated manuscript, we state our main results as Thms 3.1 and 4.1, with proofs in the proof environment, as requested.
>
> > 4. “_Since Rényi divergences are invariant under an invertible transformation of random variables [see Eq (4)], we can transform $T$ such that $\Sigma_\xi$ becomes the identity matrix without changing the information content._” Can the authors provide the proof of this?
>
> We provide a proof and a discussion of this property in the new Appx D, which we refer to from the main text (below definition 2.5).
>
> > 5. “_We note that $V\Sigma_X^{1/2}$ is orthogonal and thus $X\Sigma_XV^T$ is a diagonal matrix (Chechik et al., 2005, Lemma B.1)_”. Do you mean that by (Chechik et al., 2005, Lemma B.1) $X\Sigma_XV^T$ is a diagonal matrix, which is proved in the proof of (Chechik et al., 2005, Lemma B.1) by showing that $V\Sigma_X^{1/2}$ is orthogonal?
>
> That is indeed what we meant. We now include a proof of this property in Sec 2.2 for completeness.
>
> >  6. Eq (12) in Chechik et al. (2005) justifies writing $A$ under the form $A=WV$. Do the authors have a similar result for the Rényi IB problem that I may have missed?
>
> This form reflects the canonical transformation which is an invertible map that converts *X* and *Y* into new random variables, with a joint distribution that factorizes into a product of bivariate Gaussian distributions. We explain this important property in the new Sec 2.2.
>
> > 7. Eq (19) is confusing. Please use the notation $M^{-1}$ for the inverse of a matrix $M$ instead of a ratio $1/M$.
>
> We have changed the notation, as suggested.
>
> > 8. _“We also assume that R and W are invertible”_. Please add this assumption when stating your theorem. What happens in the case where $R$ and $W$ are not assumed to be invertible? Can you adapt the reasoning from Lemma D.1 of Chechik et al. (2005), in which they distinguish between the case where $W$ is full rank ($k=n_x$) and when it is not ($k<n_x$).
>
> We only used this assumption to simplify Eq (19) in the previous manuscript. We have updated our presentation of the proof such that this assumption is no longer required.
>
> >9. Can  the  authors  justify  why  they  seek  a  solution  of  the  form (20)? Or  is  it  an  assumption?
>
> This form results from the canonical representation in which the IB problem is equivalent to scalar IB problems for each canonical mode (the eigenmodes of the normalized regression matrix). We have reorganized our proof to clarify this point.

---

> ### Author Response · Authors · 2023-05-22
> **Author Response (3/3)**
>
> > 10. The  substitution  in (19) leading  to (21) is  not  straightforward  for  me. It  would  help  to  have  intermediate results  showing  how (19) simplifies  to (21).
>
> We now formulate our derivations in the canonical representation and replace matrix equations with much simpler scalar ones which we hope will be more transparent.
>
> > 11. _“ansatz”_ seems  to  be  a  typo.
>
> We apologize for the confusion. Our updated manuscript provides a more direct derivation without assuming that the solution takes a specific form (Eq (20) in the previous manuscript).
>
> > 12. I  have  trouble  reconciling  the  proof  made  when  $R$  and  $W$ are  invertible  with  the  following  paragraph _“Although  the  above calculation  does  not  uniquely  determine  the  mixing  matrix $W$… where  the  summations  are  restricted  to  the  eigenmodes  that contribute  the  IB  encoder, i.e., those  with  $u_i>0$”_. Under the assumption that $R$ and $W$ are invertible, shouldn’t we have that all $u_i>0$?
>
> We have reformulated our derivation so that it does not require the invertibility assumption.
>
> > 13. _“To complete our analysis of Rényi IB, we note that the analytical solution of Chechik et al. (2005) is a limiting case of our results.”_ Following my previous point, is the analysis of Rényi IB done by the authors a limiting case of Chechik et al. (2005) in the particular setting where $R$ and $W$ are assumed to be invertible?
>
> Please, see the above reply.
>
> >### 5 Jeffreys Information Bottleneck for Gaussian variables
> > 1. I believe the comments I made earlier apply to section 5 as well. For example, **3.1 Gaussian variables, 1.** for (30), **4 Rényi Information Bottleneck for Gaussian variables, 3,4,6,7** [for (32) and (33)], **8**....
>
> We have revised our analyses for both Rényi and Jeffreys cases, as suggested. Please, see our replies to the specific comments above.
>
> > 2. In the equation after (33), consider changing $A$ to something else, as $A$ is a reserved notation.
>
> We have removed this equation as it is not needed in our updated derivations.
>
> >### Appendix B
> > Consider referring to (44) as the Schur complement formula.
>
> We made the edit, as suggested.
>
> &nbsp;
>
> ### Requested Changes:
>
> > The following proposed adjustments are critical to me:
> > 1.  Add a background section according to the suggestions made for  **2 Divergence-based Correlation measure**  and  **3.1 Gaussian variables**  above.
> > 2.  Address the main points described in  **4 Rényi Information Bottleneck for Gaussian variables**  (i.e. 3,4,6,8,9,10,12,13) and reflect those changes in  **5 Jeffreys Information Bottleneck for Gaussian variables**  as well.
> >
> > Other adjustments that would strengthen the work are the remaining adjustments listed in  **Strengths And Weaknesses**  above.
>
> Please, see our response above.

---

### Review · Reviewer_4uYr · 2023-04-19

**Summary Of Contributions:**

This paper studies the information bottleneck method, defined as extracting the relevant information in a signal $X$ as the information that this signal provides about another signal $Y$. This is formulated as an optimization problem, which is optimized over encoders that map the signal $X$ to a feature representation $T$ based on a mutual information criterion.

This paper is motivated by the fact that the optimization problem corresponding to the information bottleneck method is computationally costly. As a result, the paper focuses on a special case where the source variable $X$ and the target variable $Y$ are Gaussian-correlated random variables. An exact analytical solution exists for this special case if one uses Shannon mutual information as the mutual information criterion in the optimization problem.

With this context in mind, the paper considers generalized information bottleneck methods based on Renyi divergence and Jeffreys divergence. The main result of this paper is to derive the exact analytical solution for these two divergence measures.

To derive the main result, this paper additionally considers a family of noisy linear encoders, meaning the intermediate representation satisfies a linear model given the signal $X$. Under this condition, the analytical solution satisfies a fixed-point equation that can be solved analytically from the signal $X$ and the target $Y$.

**Audience:**

Yes

**Claims And Evidence:**

No

**Requested Changes:**

**List of Requested Changes**

1. I find the linear modeling assumption from $X$ to $Y$ to be very limiting. While I understand that an extension from linear to nonlinear encoders may seem intractable, would it be possible to argue that this work is still relevant even for nonlinear encoders? I am marking No under Claims and Evidence because the paper motivates the study based on "practical interest" and "new insights into learning phenomena" of the information bottleneck method. However, I do not see why studying linear encodes under Gaussian correlated random variables tells much about either aspect.

2. Even for linear encoders with Gaussian-correlated random variables, it would be helpful to provide some illustrative experiment that applies the method in some settings.

3. Figure 1 is very cluttered. I am unable to understand the details & results that this figure is trying to show.

4. What is a compatible square matrix $C$?

5. What is an ansatz?

6. The paper's result seems to relate to Chechik et al. (2005) heavily. It would be helpful to discuss the connection more concretely with more details in related work (e.g., in the appendix).

7. Can results be provided beside the Gaussian correlated random variables of $X, Y$? While I marked Yes under Audience, the assumptions behind the main result are so strong that this paper's potential audience would probably be very limited.

**Strengths And Weaknesses:**

**List of Strengths**

S1) Well-written paper with a detailed derivation that is easy for readers to follow.

S2) The assumptions for the main result are clearly stated.

S3) Related work discussions give the reader a clear picture of this work in the perspective of the existing literature.

**List of Weaknesses**

W1) The main result assumes the encoder follows a linear relationship between the signal and the representation.

W2) Unclear what is the benefit of deriving the analytical solutions for Renyi and Jeffreys divergences rather than Shannon mutual information.

W3) No experiment and no concrete application of the theory.

W4) The main result assumes the joint distribution of $[X, Y]$ follows a normal distribution.

---

> ### Author Response · Authors · 2023-05-22
> **Author Response**
>
> Thank you for highlighting the clarity of our work and for your valuable suggestions which helped improve our paper. Please, see below our point-by-point response to your comments. (Note that we reordered some of the comments for clarity.) Please, let us know if we misunderstood any of your points.
>
> ### Requested Changes:
>
> >**List of Requested Changes**
> > 1.  I find the linear modeling assumption from  $X$  to  $Y$  to be very limiting. While I understand that an extension from linear to nonlinear encoders may seem intractable, would it be possible to argue that this work is still relevant even for nonlinear encoders? I am marking No under Claims and Evidence because the paper motivates the study based on "practical interest" and "new insights into learning phenomena" of the information bottleneck method. However, I do not see why studying linear encodes under Gaussian correlated random variables tells much about either aspect.
>
> > 7.  Can results be provided beside the Gaussian correlated random variables of  $X$, $Y$? While I marked Yes under Audience, the assumptions behind the main result are so strong that this paper's potential audience would probably be very limited.
>
> We believe that exactly solvable models are valuable in their own right as well as foundational in our understanding of more general problems. We anticipate that some of TMLR's audience will be interested in our work. In particular, our findings offer a generalization of a previous work (Chechik et al _JMLR_ 2005) that has garnered considerable interest from the community seeking to understand artificial and biological learning systems.
>
> We agree that linear encoders are somewhat limiting. However, they are an important special case that continues to offer new insights into modern machine learning of which our understanding is still lacking, see, *eg,*
> 	- Saxe et al [*Exact solutions to the nonlinear dynamics of learning in deep linear neural networks*](https://openreview.net/forum?id=_wzZwKpTDF_9C) ICLR 2014
> 	- Lucas et al [*Don't Blame the ELBO! A Linear VAE Perspective on Posterior Collapse*](https://proceedings.neurips.cc/paper/2019/hash/7e3315fe390974fcf25e44a9445bd821-Abstract.html) NeurIPS 2019
>
> In addition, we would like to point out that in many interesting settings, it is possible to control the data. The IB solution for Gaussian correlated data allows for a principled and exact investigation of the optimality and adaptability of learning systems. See, _eg,_ Palmer et al [*Predictive information in a sensory population*](https://www.pnas.org/doi/10.1073/pnas.1506855112) PNAS 2015.
>
> >2.  Even for linear encoders with Gaussian-correlated random variables, it would be helpful to provide some illustrative experiment that applies the method in some settings.
>
> We depict example applications of our results in Figs 1 and 2, where we consider a specific instance of the eigenvalues of the normalized regression matrix (as described in the figure captions).
>
> >3.  Figure 1 is very cluttered. I am unable to understand the details & results that this figure is trying to show.
>
> Thank you for pointing out the opportunity to improve our presentation. We significantly edited the figure caption for greater clarity. Below, we briefly summarize what this figure shows.
>
> This figure illustrates our solution to the Gaussian IB problem, applied to an example normalized regression matrix (described in the caption).
>
> In Fig 1a, we depict the IB frontier, *Ω(T;X) vs Ω(T;Y)*, for three different families of correlation measures: Shannon (red dotted), Jeffreys (orange dashed) and Rényi (solid, with colors corresponding to Rényi order *q*). The IB frontier provides a non-linear upper bound on *Ω(T;Y)* (i.e., how much information a representation of *X* can have about *Y*) at any given *Ω(T;X)* (i.e.,  the information a representation of *X* has about *X*). We see that this frontier depends on our choice of information measure. In addition, we show that, for all correlation measures, the IB frontiers are bounded by the data processing inequality (DPI, black dotted) and its tight, data-dependent version (SDPI, blue dashed).
>
> In Fig 1b&c, we show *Ω(T;X)* and *Ω(T;Y)* as a function of the tradeoff parameter *β*. The vertical lines are at the locations of the critical tradeoff parameters. We see no nonanalytic behavior in information at these critical points, suggesting that the transitions are continuous in all cases considered.
>
> >4.  What is a compatible square matrix  $C$?
>
> What we mean by compatible is that the dimensions of *C* are suitable for multiplication with the other matrices in an expression. In the revised manuscript, we have reformulated our derivations such that they do not rely on the matrix identity where this statement originally appeared.
>
> >5.  What is an ansatz?
>
> We apologize for the confusion. We have removed this term. Our updated analysis provides a more direct derivation that does not assume a specific form of a solution.

---

> ### Author Response · Authors · 2023-05-22
> **Author Response**
>
> _continued from previous comment_
>
> &nbsp;
>
> >6.  The paper's result seems to relate to Chechik et al. (2005) heavily. It would be helpful to discuss the connection more concretely with more details in related work (e.g., in the appendix).
>
> We have added a new background section which includes a subsection on the Gaussian IB and the results of Chechik et al (2005).
>
> > 7.  Can results be provided beside the Gaussian correlated random variables of  $X$, $Y$? While I marked Yes under Audience, the assumptions behind the main result are so strong that this paper's potential audience would probably be very limited.
>
> Please, see our response to Point 1.

---

### Review · Reviewer_oW22 · 2023-05-08

**Summary Of Contributions:**

This paper analyses the behavior of the information bottleneck problem, for Rényi and Jeffreys information (instead of Shannon), when constraining data $X$ and relevance variable $Y$ do be Gaussian and the encoder to be a noisy (Gaussian) linear map. By the use of anstaz (adding extra, unchecked, assumptions about the solution on the way), they derive a similarity of the behavior of the claimed solutions relative to the Shannon information one.



**Audience:**

No

**Broader Impact Concerns:**

For now it looks like a failed attempt at proving a theoretical result without clear insights for those who may try to solve it as well. Moreover, the assumptions (linearity and Gaussianity) are extremely restrictive in the context of state-of-the-art methods (e.g. $\beta$-VAE), which do not rely on such assumptions.

**Claims And Evidence:**

No

**Requested Changes:**

In my view, the authors still have demonstrate the usefulness of their derivation in some way before this qualifies as a finalized contribution.  In order for this work to be considered for publication I would either request:
- to write down a verifiable formal result about the studied IB problems,
- to demonstrate empirically (of course theoretically would be better, but likely hard) the performance and benefits of the proposed "ansatz" solutions.

**Strengths And Weaknesses:**

My main concern with this work is that it looks not finalized enough to be an identifiable contribution to TMLR. From the theoretical side, there is no clearly formulated theoretical result: this manifests itself by the absence of a formal proposition, which can be related to the fact that both developments (for Rényi and Jeffreys information) need to put some assumptions on the form of the solution (eq. 20 called "ansatz" in the first case, also introduced before eq. 34) to get to a conclusion. It is unclear then what to do with such conclusion: the investigated form cannot be proven to be the actual solution of the IB problem, and even from a practical perspective, it is unclear what to do with this result.

---

> ### Author Response · Authors · 2023-05-22
> **Author Response**
>
> We appreciate your valuable feedback. We take every one of your comments seriously and have addressed all of them point-by-point, making edits to our manuscript accordingly. Please, let us know if we can further clarify any of our points.
>
> ### Requested Changes:
>
> > In my view, the authors still have demonstrate the usefulness of their derivation in some way before this qualifies as a finalized contribution. In order for this work to be considered for publication I would either request:
> > - to write down a verifiable formal result about the studied IB problems,
>
> We have reorganized our manuscript and formalized our derivations. In particular, we now formulate our main results as Thms 3.1 and 4.1, complete with proofs, definitions as well as required lemmas.
>
> > - to demonstrate empirically (of course theoretically would be better, but likely hard) the performance and benefits of the proposed "ansatz" solutions.
>
> Our revised analysis does not rely on this ansatz. Our new proofs make use of the canonical representation of Gaussian correlated variables (see new Sec 2.2).
>
> We depict example applications of our results in Figs 1 and 2 for a specific instance of the eigenvalues of the normalized regression matrix (as described in the figure captions).
>
> ### Broader Impact Concerns:
>
> >For now it looks like a failed attempt at proving a theoretical result without clear insights for those who may try to solve it as well.
>
> We hope our revised manuscript which formalizes all our claims and analyses is satisfactory.
>
> >Moreover, the assumptions (linearity and Gaussianity) are extremely restrictive in the context of state-of-the-art methods (e.g.  $\beta$-VAE), which do not rely on such assumptions.
>
> (_This comment is related to that of _Reviewer 4uYr_ and we repeat some of our response here._)
>
> We believe that exactly solvable models are valuable in their own right as well as foundational in our understanding of more general problems. We anticipate that some of TMLR's audience will be interested in our work. In particular, our findings offer a generalization of a previous work (Chechik et al _JMLR_ 2005) that has garnered considerable interest from the community seeking to understand artificial and biological learning systems.
>
> We agree that linear encoders are somewhat limiting. However, they are an important special case that continues to offer new insights into modern machine learning of which our understanding is still lacking. See, *eg*,
> - Saxe et al [*Exact solutions to the nonlinear dynamics of learning in deep linear neural networks*](https://openreview.net/forum?id=_wzZwKpTDF_9C) ICLR 2014
> - Lucas et al [*Don't Blame the ELBO! A Linear VAE Perspective on Posterior Collapse*](https://proceedings.neurips.cc/paper/2019/hash/7e3315fe390974fcf25e44a9445bd821-Abstract.html) NeurIPS 2019
>
> In addition, we would like to point out that in many interesting settings, it is possible to control the data. The IB solution for Gaussian correlated data allows for a principled and exact investigation of the optimality and adaptability of learning systems. See, _eg,_ Palmer et al [*Predictive information in a sensory population*](https://www.pnas.org/doi/10.1073/pnas.1506855112) PNAS 2015.
>
> Finally, we emphasize that our main contribution is in deriving a solution to the generalized IB problem for Gaussian variables. While we believe that our work contributes to the broad area of representation learning, we do not claim that our findings are directly applicable to specific state-of-the-art methods.

---

### Author Response · Authors · 2023-05-22
**Author Response to all Reviewers**

We thank all Reviewers for their detailed feedback on our work. We have made significant edits to our manuscript to incorporate your suggestions and improve our presentation. In particular, we would like to highlight the following changes, in response to the common comments by the Reviewers.

- We have reorganized and formalized the derivations of our main results, which now appear as Thms 3.1 and 4.1, complete with proofs, definitions as well as required lemmas.
- We apologize for the confusion regarding the term _ansatz_, which we have removed. Our revised manuscript provides a more direct derivation, based on the canonical representation of Gaussian correlated variables (see new Sec 2.2). This derivation does not require this assumption on the form of a solution.

Please, see our reply for point-by-point response to other specific comments.

---

### Decision · Action_Editors · 2023-06-19

**Recommendation:** Reject

**Comment:**

Overall, this is a decent paper and technically sound. As such, the decision to reject is not a hard reject. The authors are strongly encouraged to resubmit a new version of the paper.

But, the AE also strongly recommends to the authors that they should take more time in the paper to demonstrate the relevance and articulate the contributions more clearly. How do these analytic solutions help us to understand IB approaches and representation learning more broadly? Can the authors provide any practical (even toy) demonstrations of how this can help with an ML problem? For example, the authors note that it is possible to control the data in many interesting settings. Can they provide a concrete example, even one in a relatively simple domain, with some example of the utility of their theory?

The authors are correct that there is some inherent worth in and of itself of providing analytic solutions, and using assumptions like linearity to do so is not bad, per se. But, TMLR is not a mathematics journal, and there is still a requirement to help the reader easily see why these solutions are relevant/helpful, and in this respect, the reviewers and the AE feel that the paper is still lacking, despite some attempts by the authors. As such, the decision is reject, but with a recommendation to resubmit after some additional work to make clear the contributions and relevance to other machine learning researchers.

**Audience:**

Audience relevance is the challenging aspect with this paper. The authors provide an analytic solution using some fairly restrictive assumptions, which led two reviewers to wonder about whether the results were really of interest to the TMLR audience. As well, the other reviewer felt that some of the contributions were hard to identify. After looking over the reviewer recommendations, and the paper itself, the AE feels that this paper could still use some work on making the contributions and relevance for the TMLR audience easier to identify.

**Claims And Evidence:**

This paper paper provides an analytical solution for information bottleneck (IB) approaches to representation learning, extending the work of Chechik et al. (2005) generalising Shannon information with Rényi and Jeffreys divergence. The authors rely on some assumptions (e.g. linear encoder, Gaussian correlated variables), but use these assumptions to derive an analytic solution that allows them to show some of the same phase transitions observed with earlier IB objectives. The authors suggest that this formulation of the IB problem with alternative dependence measures could offer a strategy for obtaining an approximate solution to the original IB problem and find applications in physics problems.

One of the reviewers found some of the original proofs and derivations needed clarification, but the authors have largely addressed these concerns. Thus, on a purely technical level the claims are supported.

**Resubmission Of Major Revision:**

The authors may consider submitting a major revision at a later time.